# Prevalence, Phylogroups and Antimicrobial Susceptibility of *Escherichia coli* Isolates from Food Products

**DOI:** 10.3390/antibiotics10111291

**Published:** 2021-10-22

**Authors:** Babak Pakbin, Samaneh Allahyari, Zahra Amani, Wolfram Manuel Brück, Razzagh Mahmoudi, Amir Peymani

**Affiliations:** 1Medical Microbiology Research Center, Qazvin University of Medical Sciences, Qazvin 34197-59811, Iran; b.pakbin@ut.ac.ir (B.P.); samaneh.alahyari66@gmail.com (S.A.); z.amani@ut.ac.ir (Z.A.); a.peymani@qums.ac.ir (A.P.); 2Institute for Life Technologies, University of Applied Sciences Western Switzerland Valais-Wallis, 1950 Sion, Switzerland

**Keywords:** *Escherichia coli*, antimicrobial resistance, food samples, phylogenetic group

## Abstract

The emergence of multi-drug resistant *E. coli* is an important matter of increasing considerable concern to global public health. The aim of this study was to investigate the incidence, antibiotic resistance pattern and phylogroups of *E. coli* isolates obtained from raw milk, vegetable salad and ground meat samples collected from Qazvin Province (Iran). Culture-based techniques, Kirby-Bauer disk diffusion susceptibility testing and PCR assays were used to determine the incidence rate, antimicrobial resistance pattern and phylogenetic groups of the *E. coli* isolates. The *E. coli* isolates were highly resistant to amoxicillin (79.1%), trimethoprim-sulfamethoxazole (70.8%), amoxicillin-clavulanic acid (62.5%), tetracycline (54.1%), chloramphenicol (54.1%), nitrofurantoin (54.1%), ampicillin (45.8%), streptomycin (45.8%), and kanamycin (33.3%); and completely susceptible to norfloxacin and azithromycin and 70.8% of the isolates were multi-drug resistant. Most *E. coli* isolates (46%) belonged to phylogroup A. Novel, practical, efficient food safety control and surveillance systems of multi-drug resistant foodborne pathogens are required to control the foodborne pathogen contamination.

## 1. Introduction

Foodborne diseases are defined as disorders caused by agents (bacteria, fungi, viruses, parasites and chemicals) that are usually either toxic or infectious in nature and enter the human body through the ingestion of food or drinks [1]. Foodborne bacteria are the main microbial factors causing foodborne diseases with significant adverse effects on human health and economic well-being. Foodborne pathogens can induce mild to severe both intestinal and extra-intestinal symptoms in humans [2]. The World Health Organization (WHO) has estimated that considered foodborne hazards caused more than 600 million foodborne diseases leading to more than 33 million disability-adjusted life years and 420,000 deaths annually worldwide [3]. The global burden of foodborne illnesses is comparable to those of the main infectious diseases such as tuberculosis, HIV and malaria; and certain risk factors including unimproved sanitation, air pollution and dietary risk factors [4]. At least 90% of all foodborne diseases are caused by diarrheal disease agents, indicating that many diarrheal illnesses are pathologically benign. The most prevalent foodborne pathogens contributing to the global burden of diarrheal illnesses have been reported to be noroviruses, *Campylobacter* spp., *Escherichia coli*, *Salmonella* spp., *Shigella* spp., *Giardia* spp., and *Entamoeba histolytica*. *E. coli* is one of the main challenges and concerns in food safety and public health [3].

*Escherichia coli* is a Gram-negative rod-shaped facultative anaerobic bacterium belonging to the *Enterobacteriaceae* family. *E. coli* is a commensal bacterium and typical inhabitant of the gastrointestinal tract of warm-blooded animals such as mammals, including human, cattle, and pigs, amongst others [5]. *E. coli* mostly remains confined to the human or animal gastrointestinal lumen. However, these bacteria can also cause severe intestinal and extraintestinal infections in immunocompromised or debilitated hosts [6]. Diarrheal diseases and extraintestinal infections such as hemorrhagic uremic syndrome, sepsis and meningitis in both humans and animal are caused by specific groups of *E. coli* [7]. Nevertheless, these bacteria have been generally regarded as an indicator of human fecal contamination due to their consistent presence in human feces. Contamination with either pathogenic or non-pathogenic *E. coli* may occur from animal, environmental or human sources during any of the farm-to-table processing steps and cause foodborne diseases and outbreaks in human and animals [8]. Investigation of the serogroups, pathotypes, sequence types and phylogroups in *E. coli* strains allows the characterization of potential pathogenicity and virulence of the isolates from human, animal, food, water and environments [9].

A classification system based on phylogenetic characterization of *E. coli* has previously been developed by Clermont et al. for tracking the microbial source, determination of phylogenetic groups and potential pathogenicity among the *E. coli* strains [10]. Phylogenetic analysis of *E. coli* isolates indicated that *E. coli* clones are divided into four main distinct groups, A, B2, B2 and D, and seven subgroups consisting of A0, A1, A2, B22, B23, D1 and D2. A revised phylogenetic grouping system has been proposed by Clermont et al. and four additional phylogroups including C, E, F and *Escherichia* cryptic clade I were added [11]. *E. coli* strains belonging to the phylogroup B2, and a lesser extent to the phylogroup D, are main causes of extra-intestinal infections in human. Also, the strains belonging to the phylogroup A are mostly commensal [12]. *E. coli* strains causing diarrheal diseases are most probably of the phylogroups B2 and E. There is a strong link between the virulence and phylogeny in *E. coli* infections. Phenotypic and genotypic properties of *E. coli* strains belonging to all phylogroups are entirely different. These properties include sugar fermentation, growth temperature, presence or absence of virulence encoding genes and antibiotic resistance pattern [13].

Antibiotics have a major role in treatment of bacterial infections in humans and animals by decreasing mortality and morbidity associated with the infectious diseases. Consequently, antimicrobial resistance in foodborne pathogens have increased in some countries around the world [14]. Several causative factors have been attributed to this increase including use of antibiotics for growth promotion of farm animals, addition of clinical antibiotics to the farm animal feeds, and overuse of antimicrobial agents in humans and animals [15]. The emergence of multidrug resistant (MDR) foodborne pathogens has been considered one of the main concerns in public health. MDR has been defined as acquired resistance to at least one antimicrobial agent in three or more antibiotic categories [16]. MDR *E. coli* has been recognized as one of the most significant challenges in food safety [17]. Little is known about the phylogroups of *E. coli* strains isolated from food products [18]. The purpose of this study was to determine the prevalence rate, phylogroups and antimicrobial susceptibility patterns in *E. coli* strains isolated from food products including vegetable salad, raw cow milk and ground meat samples collected from Qazvin Province (Iran).

## 2. Results

### 2.1. Isolation and Identification of E. coli in Food Products

A total of 24 (6.9%) *E. coli* strains were isolated and confirmed from 345 food samples. Prevalence rates of *E. coli* isolated from raw milk, vegetable salad, and ground meat samples collected from Qazvin Province (Iran) are illustrated in Figure 1. 13.0, 3.4 and 4.3% of the raw milk, vegetable salad and ground meat samples were contaminated with *E. coli*. All *E. coli* isolates were primarily isolated and detected using culture-based methods and then confirmed by biochemical tests. The prevalence rate of the *E. coli* isolates was significantly (*p* < 0.01) higher in raw cow milk than that in vegetable salad and ground meat samples.

### 2.2. Antimicrobial Susceptibility Evaluation the of E. coli Isolates

All twenty-four *E. coli* isolates were tested for their antimicrobial resistance against nine different common classes of antibiotics and nineteen different commercial antibiotics. The results of phenotypic resistance tests to antibiotics of the isolates are shown in Table 1. Amoxicillin (19 isolates; 79.1%), trimethoprim-sulfamethoxazole (17 isolates; 70.8%), amoxicillin-clavulanic acid (15 isolates; 62.5%), tetracycline (13 isolates; 54.1%), chloramphenicol (13 isolates; 54.1%); nitrofurantoin (13 isolates; 54.1%), ampicillin (11 isolates; 45.8%); streptomycin (11 isolates; 45.8%) and kanamycin (eight isolates; 33.3%) resistance were the dominant resistance phenotypes among the *E. coli* isolates. However, the lowest antibiotic resistance phenotypes were observed against cefoxitin (four isolates; 16.6%), colistin (four isolates; 16.6%), cefepime (three isolates; 12.5%), imipenem (one isolate; 4.1%), amikacin (one isolate; 4.1%), gentamicin (one isolate; 4.1%), nalidixic acid (one isolate; 4.1%) and levofloxacin (one isolate; 4.1%). All *E. coli* isolates were completely sensitive to norfloxacin and azithromycin. No significant differences were seen among the antibiotic resistance patterns of the *E. coli* strains isolated from raw milk, vegetable salad, and ground meat samples. 70.8% of the *E. coli* isolates expressed resistance to at least three different classes of antimicrobial agents and were considered as MDR *E. coli* isolates (Table 2). In this study, five isolates showed resistance against six antibiotic classes. 8.3% of the isolates (*n* = 2) were resistant to the β-lactam, nitroheterocyclic, aminoglycoside, folate pathway antagonist, lipopeptide, tetracycline and phenicol antibiotic categories as the most resistant profile. In summary, most of the isolates (*n* = 14; 58.3%) were resistant to four, five or six classes of antibiotics simultaneously, whereas only 4.1% (*n* = 1) and 8.3% (*n* = 2) of the isolates were simultaneously resistant to three and seven classes of antimicrobial agents, respectively.

### 2.3. Phylogroups of the E. coli Isolates

According to phylogenetic grouping of the 24 *E. coli* isolates in this study, 11 (46%), five (21%), five (21%) and three (12%) of the isolates were assigned to the phylogenetic groups of A, E, B1 and D, respectively (Figure 2). As shown in Table 3, phylogenetic groups A (seven isolates; 29.1%) and E (four isolates; 16.6%) were the most prevalent phylogroups among the *E. coli* strains isolated from raw milk samples. No D and E phylogenetic group *E. coli* strains were isolated from vegetable salad samples. Also, there are not any significant differences in the distribution of phylogenetic groupings among the *E. coli* strains isolated from each food product (Table 4). *E. coli* strains isolated from each food product were not grouped in a same phylogroups, indicating a high level of phylogenetic diversity among the strains isolated from each food product. Also, as it can be seen in Table 4, mostly diverse patterns of antibiotic resistance can be found among the isolates from each food product.

## 3. Discussion

Most of *E. coli* strains attach and harmlessly colonize the human and animal colon region of the gastrointestinal tract and only seldomly cause mild to severe intestinal and extraintestinal diseases in immunocompromised individuals [7]. Conversely, diarrheal diseases caused by pathogenic *E. coli* are a severe public health problem and concern and a major cause of mortality and morbidity, especially in children and infants [19]. Because of poor living conditions such as poor sanitation, environmental hygiene, and insufficient education, diarrheal diseases with lethal outcomes are mainly prevalent in developing and low-income countries [20]. Several studies have reported that foods of animal origin might be an important source of human-acquired MDR pathogenic *E. coli* [17]. Poultry and meat products can be widely contaminated with pathogenic or non-pathogenic groups of *E. coli* of animal origins, including MDR strains [21]. Because the main reservoir of this bacteria are the intestinal tract and feces of warm-blooded animals and humans, the presence of *E. coli* strains (pathogenic or non-pathogenic) in foods, drinks, water and other environments has been used as an indicator of fecal contamination, poor hygiene and sanitation standards during food production, processing, and distribution [22]. Livestock is considered as the main source of food contamination and the primary cause of several foodborne outbreaks due to the consumption of food contaminated with pathogenic *E. coli*. These pathogens are commonly transmitted into different foods directly and indirectly, for instance via contamination with fertilizers [20]. In many developing countries, such as Iran, animal-based fertilizers have still been used in agriculture processes and regarded as one of the most important sources of contamination of food with enteric pathogenic *E. coli* [22].

Few current studies are available regarding the prevalence rate *E. coli* strains in food products [18]. We aimed to provide determination of *E. coli* prevalence rate in food products including vegetable salad, raw milk, and ground meat samples collected from restaurants and local markets located in different areas of Qazvin Province (Iran). The total prevalence rate of *E. coli* in this study was 6.95% (24 out of 345 samples) which was higher than that reported from Mexico (1.08%; 56 out of 5162 food samples) [23], Colombia (2.1%; eight out of 380 food samples) [24], Korea (2.2%; 96 out of 4330 food samples) [25], Iran (4.0%; four out of 100 samples) [26], and Japan (6.0%; 20 out of 333 samples) [27]. The *E. coli* prevalence rate results indicate that poor hygiene and low standards of sanitation practices and behaviors have been employed during food production, processing, and distribution [27,28,29]. The results reported here also suggested that higher incidence rates of *E. coli* may necessarily imply considerable *E. coli* contamination in food in Iran [22,26,30] as we observed in this study. In this study, 54.1, 20.8, and 16.6% of all *E. coli* isolates (15, five and four out of 24 isolates) were isolated from raw cow milk, ground meat and vegetable salad samples, respectively. We found that the incidence rate of *E. coli* strains in raw milk samples was higher than that in vegetable salad and ground meat samples. Raw milk could be considered as the most potential food vehicle of transmission for *E. coli* strains in Iran and around the world [31]. A higher prevalence rate and presence of *E. coli* strains in raw milk samples is an important indicator of poor hygiene practices, sanitation and fecal contamination in raw milk production and distribution. It is fairly reasonable to assume that the main source of *E. coli* contamination could be the consequence of human fecal contamination of those raw milk products during production or distribution activities [32]. In this study, we also observed higher level of contamination with *E. coli* in raw milk samples. It is strongly recommended to consume heated milk, and thus to decrease the risk of foodborne diseases caused by enteric pathogenic *E. coli*. This also indicates poor hygiene and sanitation practices and fecal contamination during raw milk production and distribution [28].

*E. coli* isolates were highly resistant to amoxicillin, trimethoprim-sulfamethoxazole, amoxicillin-clavulanic acid, tetracycline, chloramphenicol, nitrofurantoin, ampicillin, streptomycin, and kanamycin as well as completely sensitive to norfloxacin and azithromycin antibiotics. One of the main concerns in food safety and public health is the emergence of antibiotic resistant foodborne bacterial pathogens [33]. A wide range of antimicrobial agents are currently being employed worldwide for growth promotion, diseases prevention, and treatment of sick animals allowing the development of MDR foodborne pathogens [34]. A previous study in Korea reported high level of antibiotic resistance among *E. coli* strains isolated from food and food animals, finding that 15.6, 12.5, 10.4, 9.4, and 9.4% of the *E. coli* isolates were resistant against tetracycline, streptomycin, ampicillin, ticarcillin and nalidixic acid antibiotics, respectively [25]. A study which was conducted in Mexico reported that *E. coli* strains isolated from food samples were resistant to tetracycline (34%; 19 out of 56), cefepime (30%; 17 out of 56) and ampicillin (29%; 16 out of 56) [23]. A study in Iran by Mazaheri et al. reported high resistance to tetracycline and ampicillin in Shiga toxin-producing *E. coli* (STEC) strains isolated from lettuce samples [26]. Another study which has been performed recently by Wang et al. showed that 49, 28, 24, 20 and 18% of *E. coli* strains isolated from retail food samples were resistant to tetracycline, nalidixic acid, ampicillin, sulfamethoxazole-trimethoprim and cephalothin, respectively [27]. Moreover, Yu et al. in China isolated *E. coli* from raw milk samples with high resistance to penicillin, acetylspiramycin, oxacillin, lincomycin, sulphamethoxazole, cephalosporin and ampicillin [28]. A recent study of Elmonir et al. in Egypt also showed a high resistance to nalidixic acid, ampicillin and streptomycin in STEC strains isolated from raw milk and beef samples [35].

Obviously, resistance to different classes of antimicrobial agents among the *E. coli* isolates from food products has recently increased worldwide [18,28,35]. From the regional point of view, antibiotic resistance patterns of *E. coli* isolates from different food products at the present study are partly in agreement with previous studies in Iran [26]. In this study, 70.8% (17 out of 24) of *E. coli* isolates expressed resistance to at least three different classes of antibiotics and were regarded as MDR *E. coli* strains. The MDR rates of *E. coli* isolated from food samples reported in this study was significantly higher than that reported in Korea (12.5%) [25], Turkey (20%) [36], and Egypt (51.42%) [35], and lower than those reported in China (100%) [28] and Mexico (92.4%) [23]. The continuous global resistance among *E. coli* strains of food origins has been considered a serious threat to the public health and a major food safety concern [16]. The indiscriminate and irrational use of clinical and veterinary antibiotics in the water and feed of lifestock (for infection treatment and/or growth promotion) may contribute to multidrug resistance in *E. coli* strains of food origin, which is becoming a serious public health concern [18,27,28]. This may be regarded as the main reason for high prevalence of MDR *E. coli* isolates found in the food samples of this study.

*E. coli* strains of the same phylogenetic group share similar phenotypic and genotypic characterizations, disease-causing ability, life history features and ecological attributes [37]. Different phylogenetic groups of *E. coli* have been found in specific hosts and demonstrated the same level of adaptability to the environmental conditions [11]. Of the 24 *E. coli* isolates from food samples in this study, phylogenetic group A was the most prevalent (46%; 11 out of 24) and phylogroup D was the least common. The results of our study differ from another study has previously been conducted by Higgins et al. in the USA and they show that 26, 25 and 17% of the *E. coli* isolates from animal, humans, and water were belonging to the phylogroups B1, A and D, respectively [38]. *E. coli* isolates belonging to the phylogroups B2 most often contribute to extra-intestinal diseases; however, some strains included in other phylogroups (A and B1) have been identified as causes of diarrheal diseases in humans [37].

## 4. Materials and Methods

### 4.1. Collection of Food Samples

A total number of 345 food samples including vegetable salad (*n* = 115), raw cow milk (*n* = 115), and ground meat (made from minced beef meat) samples (*n* = 115) were purchased and collected from 27 restaurants and 45 local markets located in different areas throughout the Qazvin Province (Iran), between August 2018 and February 2019. All samples were aseptically collected in sterile tubes and containers and immediately transported to the food microbiology laboratory in cool boxes with ice packs for further analysis.

### 4.2. Isolation and Identification of E. coli

*E. coli* was isolated and identified in food samples according to the method previously described by Ombarak et al. [30]. Ten mL of raw milk, 25 g of vegetable salad and 25 g of ground meat samples were mixed with either 90, or 225 mL of tryptic soy broth (TSB, Cat #105459, Merck, Darmstadt, Germany) yielding a 1:10 sample dilution, then homogenized at 400 rpm for 10 min using a Stomacher-blender BagMixer (Cat #024230, InterScience Co., Paris, France) and incubated at 37 °C for 16 h. One hundred µL of the enriched cultures were streaked onto eosin methylene blue agar plates (EMB, Cat #101347, Merck) and incubated at 37 °C for 24 h. One presumptive *E. coli* colony on each EMB agar plate (blue-black colonies with a metallic green sheen) were selected, picked and subjected to Gram-staining (negative for *E. coli*) and biochemical tests including motility (+ for *E. coli*), oxidase (− for *E. coli*), indole production (+ for *E. coli*), citrate utilization (− for *E. coli*), methyl red (+ for *E. coli*), Voges-Proskauer (− for *E. coli*), triple sugar iron (acid/acid, gas + for *E. coli*), urease (− for *E. coli*), lysine decarboxylase (+ for *E. coli*) and recommended sugar fermentation tests (glucose +, mannitol +, lactose + and mannose + for *E. coli*) (Merck). All confirmed *E. coli* isolates were stocked in TSB medium (Merck) containing 20% (v/v) glycerol (Cat #56-81-5, Sigma Chemical Company, St. Louis, MO, USA), incubated at 37 °C for 24 h and kept at −80 °C until further analysis. *Escherichia coli* ATCC 25922 (Serotype O6) a recommended reference strain for antibiotic susceptibility testing was used as positive control [39]. The control strain was activated by inoculation into TSB medium and incubation at 37 °C for 24 h.

### 4.3. Antimicrobial Resistance Testing

Antimicrobial susceptibility testing for *E. coli* isolates was performed in triplicates using a Kirby-Bauer disk diffusion assay based on the standards and interpretive criteria previously established and developed by Clinical and Laboratory Standards Institute [40]. Nineteen commercial antibiotic disks (Oxoid Ltd., Basingstoke, UK) used in this study included cefepime (FEP), 30 µg; cefoxitin (FOX), 30 µg; kanamycin (KAN), 30 µg; ampicillin (AMP), 10 µg; imipenem (IPM), 10 µg; amoxicillin (AMX), 25 µg; amoxicillin-clavulanic acid (AMC), 20/10 µg; streptomycin (SPT), 10 µg; amikacin (AMK), 30 µg; norfloxacin (NOR), 10 µg; gentamicin (GEN), 10 µg; nalidixic acid (NAL), 30 µg; levofloxacin (LVX), 5 µg; colistin (CST) 10 µg; azithromycin (AZM), 15 µg; tetracycline (TET), 30 µg; chloramphenicol (CHL), 30 µg; trimethoprim-sulfamethoxazole (SXT), 1.25/23.75 µg, and nitrofurantoin (NIT), 300 µg. The results of antibiotic resistance phenotypes were recorded and interpreted according to CLSI guidelines [40,41]. *Klebsiella pneumoniae* ATCC 700603, *Escherichia coli* ATCC 25922 and *Staphylococcus aureus* ATCC 25923 were used as the reference strains for quality control.

### 4.4. DNA Extraction

All isolates and the control strain (*E. coli* ATCC 25922) were grown on bovine heart infusion (BHI, Cat #110493, Merck) broth overnight at 37 °C. 1 mL of Phosphate Buffered Saline (PBS, Cat #524650, Merck) was mixed with 1 mL of the bacterial suspension and centrifuged at 8000× *g* for 4 min. The supernatant was removed, and the bacterial sediment was subjected to genome extraction using Sinaclon bacterial DNA extraction kit (Cat #EX6021, Sinaclon Co., Tehran, Iran) according to the manufacturer’s instruction. Quantity and quality of the extracted DNA were measured using NanoDrop 2000 spectrophotometer (ThermoFisher Scientific, Carlsbad, CA, USA). The concentrations of the extracted DNA were adjusted to 50 μg/mL with PBS prior to PCR reactions.

### 4.5. Phylogroups Determination

To determine phylogroups in the *E. coli* isolates, a triplex PCR method described by Clermont et al. was used [11]. PCR was carried out in an ABI PCR thermal cycler model 9092 (Applied Biosystems, Foster, CA, USA). Specific primers which have previously been described by Clermont et al. were used to amplify TSPE4, *yjaA* and *chuA* genes [11]. PCRs were performed in 20-µL reaction volumes containing 10 µL of PCR Master Mix kit (Cat #A190303, Ampliqon, Herlev, Denmark), 0.5 µL of each primer (2 µM/µL), 2 µL of DNA template and nuclease-free deionized water to reach the final reaction volume. The PCR reaction was performed as follows: initial denaturation step at 95 °C for 5 min, following by 35 cycles comprised 95 °C for 30 s, 59 °C for 30 s, 72 °C for 40 s; and a final extension at 72 °C for 5 min. Amplified PCR products were separated and characterized using gel electrophoresis on a 1.5% (w/v) agarose gel (Cat #9012-36-6, Sigma Chemical Company) containing DNA safe stain (Cat #S11494, Invitrogen, Paisley, UK) at 110 V for 1 h. Gels were visualized and the phylogroup patterns were recorded using a Novin-Pars Gel Documentation system (NovinPars Co., Tehran, Iran). Non-pathogenic *E. coli* ATCC 25922 containing all three genes was used as the control strain.

### 4.6. Statistical Analysis

Fisher′s exact and Chi-square tests were used to evaluate significant differences (*p* < 0.05) between the incidence rates using SPSS version 21.0.1 (IBM Corp., Armonk, NY, USA) software. All measurements were performed in triplicate.

## 5. Conclusions

In conclusion, this study investigated the prevalence rate, antimicrobial susceptibility and phylogenetic groups of *E. coli* strains isolated from food products, including raw milk, vegetable salad, and ground meat samples. Our results demonstrated that prevalence the rate of *E. coli* was higher in raw milk samples than in vegetable salad and ground meat samples. This study showed that *E. coli* isolates were highly resistant to amoxicillin, trimethoprim-sulfamethoxazole, amoxicillin-clavulanic acid, tetracycline, chloramphenicol, nitrofurantoin, ampicillin, streptomycin, and kanamycin as well as completely sensitive to norfloxacin and azithromycin antibiotics. We found that 70.8% of *E. coli* isolates were MDR to at least three classes of antimicrobial agents. Most *E. coli* isolates (46%) belonged to phylogroup A. Novel and efficient food safety control and surveillance systems and genotyping of foodborne pathogens, especially MDR strains, using standard methods such as next generation sequencing and pulsed field gel electrophoresis assays in developing and low-income countries is strongly required to control and prevent foodborne pathogen contamination and diseases. We also believe that the low number of isolates tested is an important limitation in this study.

## Figures and Tables

**Figure 1 antibiotics-10-01291-f001:**
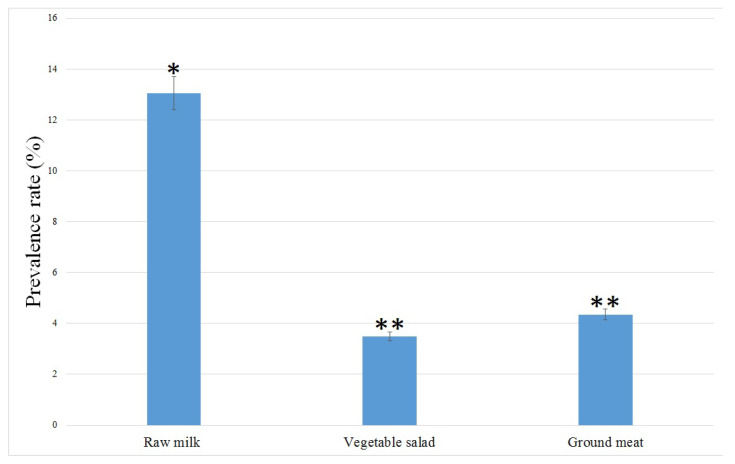
Prevalence rates of *E. coli* in different food samples collected from Qazvin province, Iran. * and ** indicates significant differences (*p* > 0.05).

**Figure 2 antibiotics-10-01291-f002:**
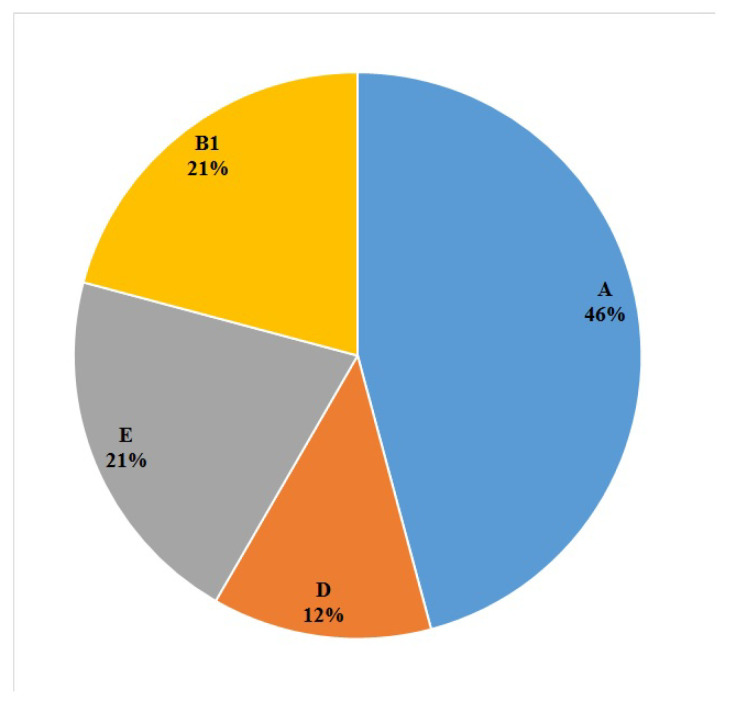
Phylogenetic group distribution of *E. coli* strains isolated from food samples.

**Table 1 antibiotics-10-01291-t001:** Antibiotic resistance phenotype of *E. coli* isolated from food samples.

Antibiotic class	Antibiotic Agent	*n* (%) ^a^			
		Raw Milk (*n* = 15)	Ground Meat (*n* = 5)	Vegetable Salad (*n* = 4)	Total (*n* = 24)
β-Lactams	Cefoxitin	3 (20.0)	0 (0)	1 (25.0)	4 (16.6)
	Imipenem	1 (6.6)	0 (0)	0 (0)	1 (4.1)
	Amoxicillin	12 (80.0)	5 (100)	2 (50.0)	19 (79.1)
	Ampicillin	8 (53.3)	2 (40.0)	1 (25.0)	11 (45.8)
	Cefepime	2 (13.3)	0 (0)	1 (25.0)	3 (12.5)
	Amoxicillin-clavulanic acid	11 (73.3)	3 (60.0)	1 (25.0)	15 (62.5)
Aminoglycosides	Streptomycin	7 (46.6)	4 (80.0)	0 (0)	11 (45.8)
	Kanamycin	8 (53.3)	0 (0)	0 (0)	8 (33.3)
	Amikacin	1 (6.6)	0 (0)	0 (0)	1 (4.1)
	Gentamicin	1 (6.6)	0 (0)	0 (0)	1 (4.1)
Quinolones and fluoroquinolones	Nalidixic acid	0 (0)	1 (20.0)	0 (0)	1 (4.1)
	Norfloxacin	0 (0)	0 (0)	0 (0)	0 (0)
	Levofloxacin	1 (6.6)	0 (0)	0 (0)	1 (4.1)
Macrolides	Azithromycin	0 (0)	0 (0)	0 (0)	0 (0)
Tetracyclines	Tetracycline	10 (66.6)	3 (60.0)	0 (0)	13 (54.1)
Lipopeptides	Colistin	4 (26.6)	0 (0)	0 (0)	4 (16.6)
Phenicols	Chloramphenicol	9 (60.0)	3 (60.0)	1 (25.0)	13 (54.1)
Nitroheterocyclics	Nitrofurantoin	10 (66.6)	0 (0)	3 (75.0)	13 (54.1)
Folate pathway antagonists	Trimethoprim-sulfamethoxazole	12 (80.0)	4 (80.0)	1 (25.0)	17 (70.8)

^a^ The number and percent of *E. coli* isolates out of the total *E. coli* isolates (*n*) in each column.

**Table 2 antibiotics-10-01291-t002:** Multidrug resistance class patterns of *E. coli* isolates (*n* = 24) from food samples.

No. Classes of Antibiotics	Multidrug Resistance Patterns ^a^ (No. Isolates in Each Pattern)	No. Total Isolates (%) ^b^ in Each Class of Antibiotic
One	βLs (*n* = 3)	3 (12.5)
Two	βLs-NHCs (*n* = 4)	4 (16.6)
Three	βLs-AGs-FPAs (*n* = 1)	1 (4.1)
Four	βLs-TCs-PNs-FPAs (*n* = 1)	4 (16.6)
	βLs-LPs-NHCs-FPAs (*n* = 1)	
	βLs-AGs-NHCs-FPAs (*n* = 1)	
	βLs-PNs-NHCs-FPAs (*n* = 1)	
Five	βLs-AGs-TCs-PNs-FPAs (*n* = 3)	5 (20.8)
	AGs-TCs-LPs-PNs-FPAs (*n* = 1)	
	βLs-AGs-TCs-NHCs-FPAs (*n* = 1)	
Six	βLs-AGs-TCs-NHCs-PNs-FPAs (*n* = 3)	5 (20.8)
	βLs-AGs-TCs-QNs-PNs-FPAs (*n* = 2)	
Seven	βLs-AGs-TCs-LPs-NHCs-PNs-FPAs (*n* = 2)	2 (8.3)

^a^ βLs-β-Lactams, NHCs-Nitroheterocyclics, AGs-Aminoglycosides, FPAs-Folate pathway antagonists, LPs-Lipopeptides, TCs-Tetracyclines, PNs-Phenicols, QNs-Quinolones and fluoroquinolones. ^b^ The number and percent of *E. coli* isolates out of the total 24 *E. coli* isolates (*n*) from all food samples.

**Table 3 antibiotics-10-01291-t003:** Distribution of phylogroups among the *E. coli* isolates obtained from different food products.

Food Product	*n* (%) of the *E. coli* Isolates (*n* = 24)			
	A	B1	D	E
Raw milk	7 (29.1)	2 (8.3)	2 (8.3)	4 (16.6)
Vegetable salad	2 (8.3)	2 (8.3)	0 (0)	0 (0)
Ground meat	2 (8.3)	1 (4.1)	1 (4.1)	1 (4.1)

**Table 4 antibiotics-10-01291-t004:** Distribution of phylogroups among the *E. coli* isolates obtained from different food products.

No.	Isolate	Food Sample	Resistance Phenotype ^a^	Phylogroup
1	ECS1	Vegetable salad	FOX, NIT	B1
2	ECS2	Raw milk	AMC, IPM, AMX, TET, CHL, SXT	B1
3	ECS3	Raw milk	AMC, KAN, AMX, TET, LVX, FEP, CHL, SXT	E
4	ECS4	Raw milk	SPT, AMC, AMX, AMP, TET, FEP, CHL, NIT, SXT	E
5	ECS5	Raw milk	FOX, AMC, KAN, AMX, TET, CHL, NIT, SXT	E
6	ECS6	Raw milk	AMC, CST, NIT, SXT	A
7	ECS7	Raw milk	KAN, TET, CST, CHL, SXT	D
8	ECS8	Vegetable salad	AMC, AMX, AMP, NIT	A
9	ECS9	Raw milk	SPT, AMC, KAN, AMX, AMP, TET, CHL, SXT	A
10	ECS10	Raw milk	FOX, SPT, AMX, AMP, GEN, NIT, SXT	A
11	ECS11	Raw milk	SPT, AMC, KAN, AMX, AMP, TET, CST, CHL, NIT SXT	A
12	ECS12	Raw milk	SPT, AMC, KAN, AMX, AMP, TET, CHL, NIT SXT	D
13	ECS13	Raw milk	FOX, AMX, NIT	A
14	ECS14	Ground meat	SPT, AMC, AMX, SXT	A
15	ECS15	Ground meat	SPT, AMC, AMX, CHL, SXT	E
16	ECS16	Ground meat	NAL, SPT, AMC, AMX, AMP, TET, CHL, SXT	D
17	ECS17	Vegetable salad	AMX, CHL, NIT, SXT	B1
18	ECS18	Raw milk	AMX	A
19	ECS19	Vegetable salad	FEP	A
20	ECS20	Ground meat	AMX	B1
21	ECS21	Raw milk	SPT, AMC, KAN, AMX, AMK, AMP, TET, CST, CHL, NIT, SXT	B1
22	ECS22	Raw milk	SPT, AMC, KAN, AMX, AMP, TET, NIT, SXT	E
23	ECS23	Ground meat	SPT, AMX, TET, CHL, SXT	A
24	ECS24	Raw milk	AMC, AMX, AMP, NIT	A

^a^ NAL-nalidixic acid, FOX-cefoxitin, SPT-streptomycin, AMC-amoxicillin-clavulanic acid, IPM-imipenem, KAN-kanamycin, AMX-amoxicillin, AMK-amikacin, AMP-ampicillin, TET-tetracycline, LVX-levofloxacin, FEP-cefepime, GEN-gentamicin, CST-colistin, CHL-chloramphenicol, SXT-trimethoprim-sulfamethoxazole, NIT-nitrofurantoin.

## Data Availability

We confirm that all data supporting the findings of this study are available within the article.

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
