# Peer review of "Prevalence, Phylogroups and Antimicrobial Susceptibility of Escherichia coli Isolates from Food Products"

_antibiotics, 2021, doi:10.3390/antibiotics10111291_

Round 1
Reviewer 1 Report
General:
The manuscript is written in good, appropriate English and clearly structured.
Nevertheless, it has some main drawbacks: The impact of the results are poor. As only low numbers of E. coli were analysed, associations between phylogroups, AMR and genotypes cannot be made.
Genotyping should be carried out by at least PFGE as gold standard. There are discrepancies in the results of phylogroups and genotypes, which are not discussed (see below). PFGE has more resolution power and for this low numbers of isolates, the costs are manageable.
References are not carefully used. I could check all, but some and found inappropriate used ones (e.g. 17(2x), 18, 21)
Details:
Abstract, Introduction and Result part: there are no information regarding the country, from which the samples originate.
Abstract: no subheadings (Background; Methods…) in the text
Introduction: line 69 – erase including
Line 71 – erase recently (8 years are not recently)
-Line 80-83: when animals and humans are treated with antimicrobials why should AMR increase only for animal related food? Hence, in some European countries, AMR in livestock even decreased within the last years (and also antimicrobial usage)
-Line 88: sentence needs consideration; messaged of the sentence not clear
Result: general just use only one decimal with this low numbers of isolates
-Line126-129: re-write the sentence; e.g “two isolates showed resistance against 7 antimicrobial classes”
- discriminating power of RAPD-PCR seems to be limited, when one cluster comprises of different phylogenetic groups; explain and discuss this; you results of the RAPD is also in contrast to your statement in discussion Line276;
-investigations of virulence genes should be done, when argue for consumers risks
Discussion:
- The study only refers to food products. However, what are the prevalence and characteristics of E. coli from food producing animals – here especially for cattle? How does agriculture works in Iran? Is manner used for fertilization? Discuss these points
-Line 185-189: sentence to long; split or erase “mostly caused by different E. coli”
-check references for use (17 and 21 are wrong)
Line 197: studies on E. coli in food are available (you mention them later); did you mean that there are only few studies, which combine human and food sector? If not, the reference is not suitable
-Line 209-212: don`t understand: your prevalence might be higher due to a good isolation method; but if you are able to isolate E. coli, than the food is contaminated; is could be that former studies underestimated prevalence due to the method used
-Line 201: erase significantly, that is not true for all countries given
-Line 219-222 Rewrite the sentence; as raw milk and Karish are almost equal contaminated on high level (something like: raw milk and raw milk cheese (Karish) was contaminated on very high levels…)
-Missing: discussion of the impact of your raw milk contamination: advice not to consume without cooking; raw milk products may pose a risk for consumers…
-paragraph form line 229 on: no relation to your results; should be introduced by sentence of line 255; Line 251-255 just repeat results; combine both paragraphs and shorten them a bit; what’s the conclusion of all these findings?
-Line 279-291: you did no investigations on virulence for your isolates. There are also a lot of examples to prove the opposite; phylogroup B2/D with high AMR but low virulence and vice versa for group A, C, B1… so this depends on your samples; if you discuss this in this way here you implicate that this might be the same for your isolates without prove
Materials: -define ground meat; different parts of the world use meat from different animal species
-how many colonies did you picked per sample? I guess one; but should be mentioned in 4.2. (did you try two pick more than one and had a look, if there might be more than one strain per sample?)
-Phylogenetic groups: you should use the refined scheme from 2013 as you mentioned the method introduction;
-Line 395: all experiments? Including phylogroup PCR or E. coli isolation?
Conclusion: too long; more an abstract character; middle part could be shorten/erase:
-Line 412-415 both points are missing in the discussion section
References: doi numbers should be given
Author Response
Dear Reviewer 1
All revisions have been made and sent to you in a tack-change form file. All changes can be found in the file.
- Association between phylogroup and AMR is removed from the manuscript.
- As it is strongly recommended, the RAPD genotyping is not suitable and precise to be implemented and used in this study and it is better to remove it from the manuscript. However, there is not any remining budget so it was not possible for us to perform any genotyping again by the gold standard methods such as PFGE and WGS techniques. Consequently, we removed this part of the study and restructured the manuscript and focused strongly on other parts including the phylogroup typing and AMR properties of the E. coli isolates from food samples. We tried our best to present these data as well according to your comments.
- All references (e.g. 17, 18, 21) are replaced with a suitable ones.
Details
- The information regarding the country from which the samples originate are added into the abstract, introduction and results sections.
- The subheadings in the abstract are omitted from the text.
- Line 69: “including” is removed.
- Line 71: “recently” is removed.
- Line 80-83: the animal related foods are removed from the text and the sentence is revised. Around the world is changed to some countries as there are some developing countries with the challenge of AMR in livestock.
- Line 88: the sentence is reviewed and revised to be clear.
- All data throughout the manuscript are adjusted to only one decimal.
- Line 126-129: the sentence is rewritten and revised in the text.
- RAPD section is completely removed from this study so the discriminating power is not considered in this study now.
- As it is mentioned in the text, in this study we used phylogrouping which is strongly associated with the virulence of E. coli isolates and describe the isolates accordingly.
- This explanation is added to the manuscript to discuss these points (The study only refers to food products. However, what are the prevalence and characteristics of E. coli from food producing animals – here especially for cattle? How does agriculture works in Iran? Is manner used for fertilization? Discuss these points): “Livestock is considered as the main source of food contamination and cause of primari-ly several foodborne outbreaks due to the consumption of foodstuff contaminated with pathogenic E. coli. These pathogens commonly are transmitted into different foodstuff directly and indirectly for instance via contamination with fertilizers. In many de-veloping countries, such as Iran, animal-based fertilizers have still been used in agri-culture processes and regarded as one of the most important sources of contamination of foodstuff with enteric pathogenic E. coli”.
- Line 185 – 189: “mostly caused by different E. coli” is removed from the text.
- The references 17 and 21 is replaced and revised.
- Line 197: The reference is changed, replaced and revised according to the text in this section.
- Line 209 – 212: as this reason is not reasonable it is removed and revised in the text.
- Line 201: “significantly” is removed from the text.
- Line 219 – 222: this sentence is rewritten and revised in the text accordingly.
- Discussion of the impact of our raw milk contamination and the advice to not consume without cooking and raw milk products are added and revised in the text.
- Paragraph from line 229: this section is revised and rewritten in the text accordingly. The repetitive parts are removed and focused on the findings in this study.
- Line 279 – 291: this section for phylogrouping is completely revised in the text according to your comments.
- The ground meat is defined and revised in the text.
- One colony was picked per sample and investigated. This explanation is added into the text.
- The refined scheme from 2013 for phylogrouping was used in this study. This statement is added and revised in the text.
- Line 395: Except isolation and phylogrouping, other measurements were performed in triplicates. Consequently, it is revised in the materials and methods section accordingly.
- The conclusion section is shortened and revised in the manuscript.
- Line 412-415: these points are added in the discussion section.
- Doi numbers are given to all references in this manuscript.
Kind regards,
Dr. Wolfram Bruck
Reviewer 2 Report
The Authors proposed an interesting study in Prevalence, Phylogroups, Antimicrobial Susceptibility and Genetic Diversity of E. coli isolated from food. The introduction described exhaustively the aim of research and the research methodology was detailed, but there were some points which I suggest revising:
Line 69: can the authors review the sentence (clone groups)
line 88-89: May Authors insert, already in the introduction, a complete definition of MDR?
Multidrug-resistant, extensively drug-resistant and pandrug-resistant bacteria: an international expert proposal for interim standard definitions for-resistant bacteria: an international expert proposal for interim standard definitions for acquired resistance A.-P. Magiorakos et al. Bacteriology – 2011
https://www.clinicalmicrobiologyandinfection.com/article/S1198-743X(14)61632-3/pdf
The results were stated clearly, but I suggest correcting “com” line 103 in “cow”.
The discussion and conclusion were well argued
Author Response
Dear Reviewer 2
As it is suggested by the reviewers, the RAPD genotyping is not suitable and precise to be implemented and used in this study and it is better to remove it from the manuscript. However, there is not any remining budget so it is not possible for us to perform any genotyping again by the gold standard methods such as PFGE and WGS techniques. Consequently, we removed this part of the study and restructured the manuscript and focused strongly on other parts including the phylogroup typing and AMR properties of the E. coli isolates from food samples. We tried our best to present these data as well according to your comments.
- Line 69: the sentence is reviewed and revised in the text.
- Line 88 – 89: A complete definition of MDR is added using the reference A. P. Magiorakos et al. 2011
- Line 103: “com” is revised to “cow”.
Kind regards,
Dr. Wolfram Bruck
Reviewer 3 Report
The manuscript provided by Pakbin et al. describes the prevalence of antibiotic resistant Escherichia coli in raw cow milk, vegetable salad and ground meat samples from 27 restaurants and 45 local markets located in different areas throughout the Qazvin province, Iran. The authors used different methods for detection and determination of E. coli isolates to different phylogroups and their genetic diversity. They confirmed the high percent of bacterial prevalence rate in raw milk than in vegetable salad and ground meat samples, which indicates poor hygiene and sanitation practices in the restaurants and markets. Most of the E. coli isolates are multi-drug resistant. The study is interesting, but I have some recommendations to authors.
I recommend that the conclusions in the abstract coincide with those in the manuscript. Also add from where the samples were taken.
The keywords should be with small caps and listed with commas.
Line 38: Write the full name of WHO.
Line 47: Escherichia coli should be in Italic.
Line 50: ”gram-negative” should be with capitalize letter.
Line 51: Enterobacteriaceae is a family and should be in Italic.
Line 40 and line 56: I do not recommend the use of abbreviations DALYs and HUS because you do not use them anywhere below in the text.
Lines 99-101: The sentence is unclear. Please rewrite it, because it is not understood in this way and the results do not match those in Figure 1.
Lines 112-125 and lines 129-130: Unify the writing of the results.
Line 126: You have already entered an abbreviation for “multidrug resistance”.
Line 133: Table 1 is not clear. You should give legend under table with clarification.
Line 134: Under the “No. total isolates (%) (n=24)” in Table 2 you give as result “3 (12.5)”. You need to clarify this.
Lines 135-136, lines 177-179: The abbreviation should be in Legend. Please list with commas and small caps. For example: βLs - β-lactams, NHCs – nitroheterocyclics, AGs – aminoglycosides, etc.
In Material and methods: Give the catalog numbers and the city of production of the materials used.
Lines 329-331: You mentioned the biochemical tests. Please, list what the isolated strains must be in order to belong to a species E. coli (negative or positive).
Line 332: I have a question. Is not the percentage of glycerol used as a cryoprotectant low? Usually 25% glycerol is used.
Lines 341-347: There is no information about antibiotics here (company, city and country of production).
The Discussion and Conclusions are well described. I will recommend your manuscript for publication after making the corrections.
Author Response
Dear Reviewer 3,
As it is suggested by the reviewers, the RAPD genotyping is not suitable and precise to be implemented and used in this study and it is better to remove it from the manuscript. However, there is not any remining budget so it is not possible for us to perform any genotyping again by the gold standard methods such as PFGE and WGS techniques. Consequently, we removed this part of the study and restructured the manuscript and focused strongly on other parts including the phylogroup typing and AMR properties of the E. coli isolates from food samples. We tried our best to present these data as well according to your comments.
- As you recommended, the conclusions in the abstract are adjusted with those in the manuscript. Also, the city and country of the samples originate are added in abstract, introduction and results sections.
- The keywords are rewritten with small caps and listed with commas.
- Line 38: full name of WHO is added.
- Line 47: Escherichia coli is revised to italic form.
- Line 50: “gram-negative” is rewritten with capitalize letter.
- Line 51: Enteroobacteriacaea is rewritten in italic form.
- Lines 40 and 56: abbreviations DALYs and HUS are removed from the text.
- Line 99-101: the sentence is rewritten and revised in the text according to the figure 1.
- Lines 112 – 125 and 129-130: the writing of results is unified and revised in the text.
- Line 126: This sentence is rewritten and revised completely.
- Line 133: Legend are added into the Table 1 to be clear.
- Line 134: new legend is added to the Table to be clear.
- Lines 135-136 and 177-179: all abbreviations are revised according to your comment.
- Catalog numbers and the city of production of the materials are added throughout the materials and methods section of the manuscript.
- Lines 329-331: The results of the biochemical tests to confirm E. coli isolate are added into the text.
- Line 332: We used 20% glycerol (revised in the text) to stock the strains as previously recommended by other researchers the glycerol percentage between 20 – 25%.
- Lines 341-347: the information abouts antibiotics including company, city and the country of the producer are added into the text.
Kind regards,
Dr. Wolfram Bruck
Round 2
Reviewer 1 Report
Removing all phylogenetic parts resulted in low impact of the study. It might be better to seach for cooporation partners, if there is no money for further analysis (althoug the costs for 2 PFGE gels are not more than 20 dollars); To draw conclusions from your results, more genetic insides into the strains and/or extend of isolates would be needed.
Abtract: you missed to update the method part
Introduction: Line 96-98: there a a lot of studies on AMR E. coli isoltes from food
Results: Table 1 and 2: why foot notes for the first isolate in each table? Reader should be able to understand, how % are calculated
Discussion: Discussion for teh little amount of result is too long; large parts have more the carachter of a review than discussing the results (e.g. line 249: what would be an explanation for high contaminated milk)
Line 222-228: food instead of foodstuff; food is used throughout the rest of the manuscript;
Line 238-242: with your discussion you also indicate, that Finnland has a poor hygiene standard; which is definetly not the case. In general it would be better use representative data (monitoring reports) for this kind of discussion. Line 241/242: why not; what is the conclusion from your results?
Refrence 26 before reference 23;
Line 329: erase "most" causality is not given; extra.intestinal deseases most often by B2 but not most B2 cause illness
Line 334-336: as long as you have not proven that by investigating virulence factors, you can`t say;
Conclusion: " a cognitive scientific apporach..." this sentence is just an empty phrase (same in abstract)
Author Response
Dear Reviewer 1
Thank you very much for your considerations and comments.
- Regarding the fact that there are strongly limited studies on phylogroups of E. coli isolates from food samples, the novelty and impact of this study may be still desirable; on the other hand, we all believe that the low number of isolates is a limitation of this study and we mentioned it in the last sentence of the conclusion in this study. As the experiments and sampling in study was implemented in Iran, it was not possible for us to search for new corporation partners and addition budget for PFGE; however, we suggested using NGS and PFGE as genotyping methods in conclusion section of the manuscript.
- Abstract: the method part is updated.
- Introduction: Line 96-98: the sentence is revised according to your comment.
- Discussion: the discussion section is shortened according to your comment.
- Line 222-228: Food is replaced with foodstuff throughout the manuscript.
- Line 238-242: this section is also revised and the conclusion from this study is added to the manuscript accordingly.
- Reference 26 is revised to 22.
- Line 329: this section is partly removed and revised in the text.
- Line 334-336: this section is also removed from the text.
- Conclusion: the sentence “ a cognitive scientific approach …” is removed and revised in this section.
Kind regards,